# A QUBO formulation for top-$\tau$ eigencentrality nodes

**Prosper D. Akrobotu**[1,2], **Tamsin E. James**[2], **Christian F. A. Negre**[3], **Susan M. Mniszewski**[2] *

**1** Department of Mathematical Sciences, The University of Texas at Dallas, Richardson, TX, United States of America, **2** Computer, Computational, and Statistical Sciences Division, Los Alamos National Laboratory, Los Alamos, NM, United States of America, **3** Theoretical Division, Los Alamos National Laboratory, Los Alamos, NM, United States of America

☯ These authors contributed equally to this work.

* smm@lanl.gov

**Data Availability Statement:** All relevant data are within the manuscript.

**Funding:** This research was supported by the U.S. Department of Energy (DOE) National Nuclear Security Administration (NNSA) Advanced

## Abstract

The efficient calculation of the centrality or "hierarchy" of nodes in a network has gained great relevance in recent years due to the generation of large amounts of data. The eigenvector centrality (aka eigencentrality) is quickly becoming a good metric for centrality due to both its simplicity and fidelity. In this work we lay the foundations for solving the eigencentrality problem of ranking the importance of the nodes of a network with scores from the eigenvector of the network, using quantum computational paradigms such as quantum annealing and gate-based quantum computing. The problem is reformulated as a quadratic unconstrained binary optimization (QUBO) that can be solved on both quantum architectures. The results focus on correctly identifying a given number of the most important nodes in numerous networks given by the sparse vector solution of our QUBO formulation of the problem of identifying the top-$\tau$ highest eigencentrality nodes in a network on both the D-Wave and IBM quantum computers.

## Introduction

There are several centrality measures used to identify the most influential node(s) within a network, each having their own benefits dependent on the data at hand or the results desired.

For example, degree centrality [1], which is based purely on the number of connections a node has, could be used for identifying the most popular person within a group of people on a social media platform (number of followers). Closeness centrality [2] is dependent on the length of the paths from one node to all other nodes in a network, prioritizing nodes that are "closer" to all other nodes as more central. This has been used for predicting enzyme catalytic residues from topological descriptions of protein structures [3]. Betweenness centrality [4], is based on the number of times a node appears when two other nodes are connected by their shortest path. This measure is often used in biological networks, for example identifying a specific protein that is important for information flow within a network, which could be used in drug discovery [5]. Katz centrality [6], measures the importance of a node through its

Simulation and Computing (ASC) program at Los Alamos National Laboratory (LANL). This research has been funded by the LANL Laboratory Directed Research and Development (LDRD) under project number 20200056DR.and ASC program. SMM, CFAN, and PDA were funded by LANL LDRD. TEJ was funded by the U.S. Department of Energy (DOE) through a quantum computing program sponsored by the Los Alamos National Laboratory (LANL) Information Science & Technology Institute. Assigned: Los Alamos Unclassified Report LA-UR-21-24030. LANL is operated by Triad National Security, LLC, for the National Nuclear Security Administration of U.S. Department of Energy (Contract No. 89233218NCA000001). The funders had no role in study design, data collection and analysis, decision to publish, or preparation of the manuscript.

**Competing interests:** The authors have declared that no competing interest exist.

immediate connections, and also the connections of other nodes through the immediate neighbors. Katz centrality has also been used within a biological setting, such as identifying disease genes [7]. PageRank centrality [8] is a variant of eigenvector centrality designed to rank web pages by importance based on links between pages or articles. This differs from eigenvector centrality as it takes into account directions between nodes (clicking from one web page to another).

It is worth noting at this point that the rankings generated by these different centrality measures are correlated, especially for the most highly ranked nodes [9–13]. Our research is concentrated on a study of the eigenvector centrality (aka eigencentrality) (EC) measure [14], and using this to determine a fraction of the most important nodes in a given network. EC has been applied in many different fields of science. For example, identifying the most important amino acid residues in proteins undergoing an allosteric mechanism [15] and predicting flow-paths in porous materials [16]. It is also relevant to current world issues related to the COVID-19 pandemic, such as identifying people deemed as "super-spreaders" and areas that are hotspots in a pandemic (network of people) [17], and also in the analysis of production chains in the financial market, where micro-sectors are identified as important nodes within a chain [18].

EC is a centrality measure which assigns to each node a value that is proportional to the sum of the values for the node's neighbors [15, 19]. With this measure, the most important node is a node that is connected to a majority of the other important nodes in the network. The scheme designed to implement the EC measure uses the entries of the principal eigenvector of the network's adjacency matrix to score the nodes in the network. The scheme is justified by the Perron-Frobenious theorem [20] which states that there is a unique largest real eigenvalue for a non-negative square matrix (here given by the adjacency matrix), with an eigenvector solution consisting of positive elements. The adjacency matrix $A = [a_{ij}]_{n \times n}$ of a network or graph $G = (V, E)$ of $n$ nodes is a square matrix that describes the network's connectivity and with entries defined by $a_{ij} = 1$ if $\{u_i, u_j\} \in E$ is an edge in $G$ for each pair $u_i, u_j \in V$ in the set of nodes of $G$ and $a_{ij} = 0$ otherwise. Therefore the EC measure assigns to each node the value

$$x_i = \frac{1}{\lambda_1} \sum_{j=1}^{n} a_{ij} x_j \ \ \forall i = 1, 2, \ldots, n \tag{1}$$

where $x_i$ is the given centrality value for the i$^{\text{th}}$ node. It is represented in matrix form by

$$A\boldsymbol{x} = \lambda_1 \boldsymbol{x} \tag{2}$$

The degree centrality can also be defined as the count of the number of walks of length one that reach the node for which centrality is being computed. EC instead, is a count of the number of walks of infinite length [9, 14]. A brief demonstration of these concepts are shown in the S1 Appendix with details also found in [10]. This idea of EC counting walks will be essential in our formulation.

As networks become larger and more complicated, the efficient calculation of the centrality or "hierarchy" of nodes in a network become more relevant in network analysis for decision making. Decision makers are usually interested in determining just a fraction of the most central nodes in the network due to limited resources or budgetary constraints. For example, in the COVID-19 pandemic, the Centers for Disease Control and Prevention (CDC), limited by inadequate vaccines, might be interested in identifying only a fraction of the population that greatly influence the spread of disease or safe spots to quarantine people. Thus, identifying the top-$\tau$ most influential nodes is an essential inevitable problem to address in network analysis. The world is likely to embrace quantum computing in the near future and hence

understanding and developing quantum computing formulations of this problem considering EC of nodes would be a major advance in the field of quantum computing. Our research, thus, seeks to address this problem by reformulating the problem of identifying the top-$\tau$ most influential nodes (highest EC nodes) as a quadratic unconstrained binary optimization (QUBO) problem to be solved on quantum computers. We now give a brief introduction to quantum computing architectures.

The steady progress in the field of quantum computing since its proposal in the early 1980s by Richard Feynman has seen researchers trying different directions to circumvent the complexity of constructing portable physical quantum computers. Currently, there are two major approaches for building quantum computers: gate-based and quantum annealing. The gate-based quantum computers are designed using quantum circuits with control and manipulative power over the evolution of quantum states to tackle general problems arising in nature [21, 22]. The quantum annealing approach, however, uses the natural evolution of quantum states to tackle specific problems such as probabilistic sampling and combinatorial optimization problems [22].

The D-Wave quantum computer is a quantum computing platform that uses quantum annealing (QA), a heuristic search method that makes use of quantum tunneling and quantum entanglement to solve the ground state of the Ising model equivalence of combinatorial optimization problems. That is, any problem to be solved by the D-Wave quantum annealer, is to be modeled as a search for the minimum energy of the Ising Hamiltonian energy function as follows.

$$E(\boldsymbol{s}) = \sum_i h_i s_i + \sum_{i<j} J_{i,j} s_i s_j \tag{3}$$

where $s_i \in \{-1, 1\}$ are magnetic spin variables subject to local fields $h_i$ and nearest neighbor interactions with coupling strength $J_{ij}$ or its Boolean equivalence obtained from the transformation $\boldsymbol{s} = 2\boldsymbol{x} - \boldsymbol{1}$ where the entries of the vector $\boldsymbol{x}$ represent the binary variables $x_i \in \{0, 1\}$ and $\boldsymbol{1}$ is a vector of ones. The Boolean equivalence of the Ising problem is referred to as a QUBO problem, with the following equation

$$E(\boldsymbol{x}) = \boldsymbol{x}^T Q \boldsymbol{x} = \sum_{i=1}^{N} x_i Q_{ii} + \sum_{i \neq j} x_i x_j Q_{ij}. \tag{4}$$

D-Wave quantum annealers include the current 2000Q and the new `Advantage` [23]. The quantum processing unit (QPU) of the D-Wave 2000Q has up to 2048 qubits and 6061 couplers sparsely connected as a Chimera graph $C_{16}$ [23]. While the `Advantage` consists of more than 5000 superconducting qubits connected with 35,000 couplers on a Pegasus graph [23]. The sparse connectivity of the chimera graph of the `D-Wave 2000Q` requires a "minor embedding" of the Ising model connectivity of the problem onto the hardware. This results in chains of physical qubits representing logical qubits leading to a maximum capacity of 64 fully connected logical qubits/variables [24]. The D-Wave quantum computers possess the ability to sample degenerate ground state solutions and have been utilized in solving several problems such as quantum isomer search [25], graph partitioning [26], community detection [27], binary clustering [28], graph isomorphism [29] and machine learning [30, 31]. It has also been used in solving physical problems related to atomistic configuration stability [32], job-shop scheduling [33], airport and air traffic management [34, 35].

The gate-based quantum computers use unitary operations defined on a quantum circuit to transform input data into a desired output data [21, 36]. This computational mechanism is employed in the design of IBM quantum computers which are made available through a

cloud-based platform called IBM Quantum (IBM-Q) Experience [37]. The unitary operations are designed to process data with high fidelity and to tackle both combinatorial optimization problems and non-combinatorial problems like prime factorization [21]. IBM-Q consists of both quantum hardware and simulators. The basic steps to follow in carrying out any experiment on the IBM-Q Experience requires first to specify a quantum circuit via a graphic interface called composer or a text-based editor (the cloud version is called quantum lab), then run the circuit on a simulator to verify specifications, and finally execute the circuit on the quantum processor for a number $N$ of shots with $N = 8192$ being the maximum allowed on current devices [36].

The main goal of this paper as stated earlier is to develop a QUBO formulation for the problem of identifying the top-$\tau$ highest EC nodes in a network and implement it on quantum annealing and gate-based quantum computers. Since EC is also a good measure for generating node ranks we also explore the possibility of our formulation generating node ordering.

The paper is organized as follows. In the next section, we present a construction of a QUBO formulation for identifying a fraction of the most important nodes of the graph by the EC measure to be implemented on quantum computing devices. We then present the software tools, implementations and results obtained from the classical and quantum computers such as the D-Wave 2000Q and IBM-Q. We follow this up with a definition of node ranks from the results obtained from solving the QUBO on the D-Wave 2000Q and provide discussions on the conclusions derived from the results. Finally, we provide a summary of the results and conclude with suggestions of future directions to be considered.

## Methods

In this section we present the mathematical formulation of the EC problem as an optimization problem. A careful examination of the scheme in Eqs (1) and (2) shows that the EC is a problem of determining the eigenvector corresponding to the leading eigenvalue of the adjacency matrix of the network. To this end, we recall some useful properties for the leading eigenvalue of a symmetric matrix.

- $\frac{x^T A x}{x^T x} \leq \lambda_1 \leq d_{max}$, where $d_{max}$ is the maximum degree of the graph.

Hence it is understood that the maximum of the set of numbers $\{\frac{x^T A x}{x^T x}\}$ coincides with the leading eigenvalue of the adjacency matrix and thus one can construct a maximization problem from this property as follows:

$$\max_{x \in \mathbb{R}^n} x^T A x \text{ s.t. } \|x\| = 1 \tag{5}$$

This maximization problem is a constrained optimization problem which is equivalent to the following unconstrained minimization problem:

$$\min_{x \in \mathbb{R}^n} \left[ -x^T A x + P \left( \sum_{i=1}^n x_i^2 - 1 \right)^2 \right] \tag{6}$$

where $P$ is the Lagrange multiplier or simply a penalty constant and we have used the fact that $1 = \|x\|^2 = \sum_{i=1}^n x_i^2$. The goal here is that the argument of a typical solution to the unconstrained minimization problem should preserve the ranking on the nodes in the network when the usual iterative scheme Eq (2) is used in determining the importance of a node. That is, we are more interested in the rank assigned than the centrality values assigned to each node in the network.

## Top-$\tau$ most influential nodes as a QUBO problem

Recall that EC measures the influence of a node in a network. We therefore seek a QUBO problem whose solution classifies the most influential nodes in a network when the EC measure is used. That is to say, the proposed QUBO should directly determine the most influential nodes that would have been identified from the eigenvector in Eq (1).

Setting up a QUBO problem requires restricting the real search space to a binary search space. To accomplish this, we split the set of nodes into two categories: most central and least central, where a value of 1 denotes a node is most central and a value of 0 denotes a node is least central. To this end, we define $\tau$ as the number of most central nodes we wish to be identified from the set of nodes (i.e. how many nodes are to be assigned the value of 1). The problem of splitting into two categories is binary as we only have two categories: 0 and 1, or high and low. The value $\tau$ therefore must be chosen with consideration of factors such as the size of the network and how many most important/influential nodes you wish to identify. This definition will require a slight modification to our unconstrained minimization model, Eq (6) and a need for more constraints. Consider the second term in Eq (6), since the search field is now binary, we have for each $i$, $x_i^2 = x_i$ and based on the definition of $\tau$, $\sum_{i=1}^{n} x_i = \tau$, we adapt the following modification to the penalty term.

$$P\left(\sum_{i=1}^{n} x_i^2 - 1\right)^2 \mapsto P\left(\sum_{i=1}^{n} x_i - \tau\right)^2 \tag{7}$$

We now write the modified penalty term in matrix notation.

$$\begin{aligned}
\left(\sum_{i=1}^{n} x_i - \tau\right)^2 &= \left(\sum_{i=1}^{n} x_i\right)^2 - 2\tau \sum_{i=1}^{n} x_i + \tau^2 \\
&= \sum_{i=1}^{n} x_i^2 + 2\sum_{i<j} x_i x_j - 2\tau \sum_{i=1}^{n} x_i + \tau^2 \\
&= (1 - 2\tau)\sum_{i=1}^{n} x_i^2 + 2\sum_{i<j} x_i x_j + \tau^2 \\
&= \boldsymbol{x}^T[(1 - 2\tau)I + U]\boldsymbol{x} + \tau^2 \\
&= \boldsymbol{x}^T C\boldsymbol{x} + \tau^2
\end{aligned} \tag{8}$$

where we have used the fact that $x_i = x_i^2$ and $C = (1 - 2\tau)I + U$, $I$ is the $n \times n$ identity matrix and $U = [u_{ij}]$ is an $n \times n$ matrix with entries $u_{ij}$ defined by

$$u_{ij} = \begin{cases} 1 & i \neq j \\ 0 & \text{otherwise} \end{cases} \tag{9}$$

We now attempt to build a problem Hamiltonian from the eigenvector equation, Eq (2). Note that

$$\begin{aligned}
(A\boldsymbol{x} - \lambda_1 \boldsymbol{x})^2 &= [(A - \lambda_1 I)\boldsymbol{x}]^T[(A - \lambda_1 I)\boldsymbol{x}] \\
&= \boldsymbol{x}^T(A - \lambda_1 I)^T(A - \lambda_1 I)\boldsymbol{x} \\
&= \boldsymbol{x}^T A^T A\boldsymbol{x} - 2\lambda_1 \boldsymbol{x}^T A\boldsymbol{x} + \lambda_1^2 \boldsymbol{x}^T\boldsymbol{x} \\
&= \boldsymbol{x}^T A^T A\boldsymbol{x} - 2d_{max}\boldsymbol{x}^T A\boldsymbol{x} + d_{max}^2 \boldsymbol{x}^T\boldsymbol{x} + \text{error} \\
&= \boldsymbol{x}^T A^2 \boldsymbol{x} - 2d_{max}\boldsymbol{x}^T A\boldsymbol{x} + d_{max}^2 \boldsymbol{x}^T\boldsymbol{x} + \text{error}
\end{aligned} \tag{10}$$

where we have used $A^T = A$ for undirected graphs. We wish to obtain the ground state of Eq (10) and determine whether there is any meaningful information leading to the identification of the most influential node of the graph. Our investigations showed no conclusive information embedded in the ground state for identifying the most central node. However, conclusive information could be draw from the first excited state. This then suggested a need to modify the above objective function Eq (10). Therefore motivated by the fact that EC is also a measure of walks of infinite length, we proposed a modified objective function whose symmetric matrix is defined by Eq (11).

$$Q = -P_0 A^2 \hat{d} \hat{d}^T A - P_0 A \hat{d} \hat{d}^T A^2 + P_1 C \qquad (11)$$

where $P_0$, $P_1$ are penalty constants such that $P_1 > P_0$, and

$$d = \sum_i^n d_i e_i \text{ where } d_i = \sum_{j=1}^n e_i^T A e_j, \text{ and } \hat{d} = \frac{d}{\|d\|} \qquad (12)$$

Here the vectors $e_i \in \mathbb{R}^n$ are the canonical basis vectors of $\mathbb{R}^n$. The form of our problem Hamiltonian $Q$ was chosen to mimic the search Hamiltonian, $H = -\gamma L - ww^T = \gamma(A - D) - ww^T$, for quantum search by a continuous time quantum walk algorithm for a marked node $w$ in a graph described by Childs and Goldstone in [38]. Here, the matrix $L$ represents the graph's Laplacian which is the difference between the graph's adjacency matrix $A$ and the diagonal matrix $D$ of degrees of the nodes. The principle behind this quantum search is that the evolution of the Hamiltonian $H$ in a time inversely proportional to the energy gap $(E_1 - E_0)$ between the ground and first excited state energy $E_0$ and $E_1$ respectively generates a rotation between a start state $s = \frac{1}{\sqrt{N}} \sum_{i=1}^n e_i$, a superposition of all the states, and a state with a significant overlap with the marked state $w$ provided the first excited state has a significant overlap with the states $s$ and $w$ over $\gamma \in (0, \infty)$ [38]. Observing that the first excited state of the minimization problem of the objective function Eq (10) always had a significant overlap with our desired solution of the top-$\tau$ most influential nodes, we examined each term of Eq (10) and constructed a symmetric matrix from the outer product of the vecotrs $A^2 d$ and $A d$ to get $A^2 d(A d)^T + A d(A^2 d)^T = A^2 dd^T A + A dd^T A^2$. The vector $d$ is introduced because of the factor $d_{max}$ in Eq (10). We proceed by solving for and examining the ground state of the QUBO problem

$$\min_{x \in \{0,1\}^n} x^T Q x$$

$$Q = -P_0 A^2 \hat{d} \hat{d}^T A - P_0 A \hat{d} \hat{d}^T A^2 + P_1 C$$

$$C = (1 - 2\tau)I + U \qquad \qquad (P)$$

$$P_1 > P_0 > 0$$

with $\tau = 1$, where the matrix $U$ is defined in Eq (9).

## Results

### Tools and implementation

Using Python packages `NumPy` [39], `SciPy` [40], `D-Wave Ocean` [41], `IBM Qiskit` [42], `NetworkX` [43], and `Matplotlib` [44] and the D-Wave 2000Q and IBM-Q hardware, we performed several experiments (with $\tau \geq 1$) on the D-Wave 2000Q/IBM-Q machines for different graphs (see Figs 1 and 2). We observed that the solution to the QUBO problem P strongly correlates with the top-$\tau$ most important EC nodes when compared to the solution

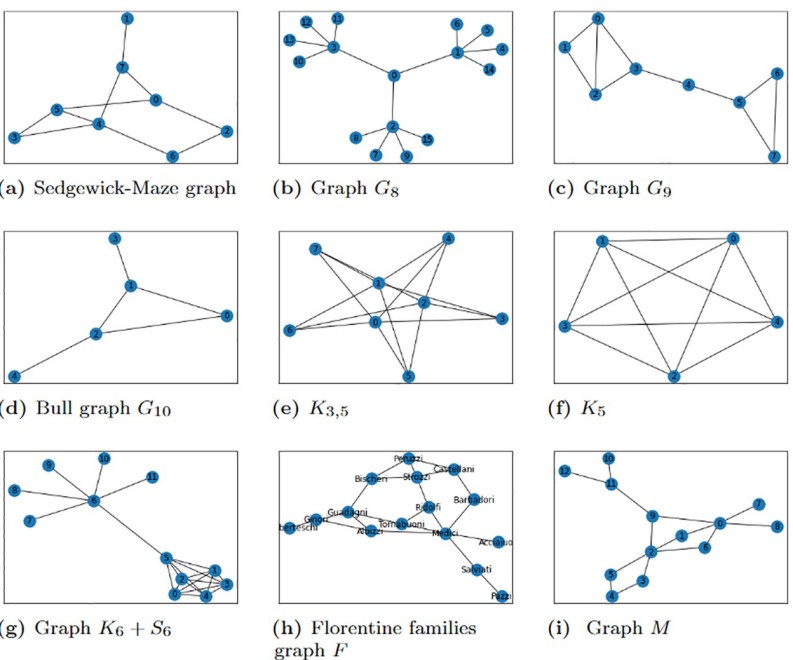

**Fig 1. Small graphs considered.** (a) Sedgewick-Maze graph, (b) Graph $G_8$, (c) Graph $G_9$, (d) Bull graph $G_{10}$, (e) $K_{3,5}$, (f) $K_5$, (g) $K_6+ S_6$, (h) Florentine families graph $F$, (i) Graph $M$.

from the power method of NetworkX's EC algorithm. The QUBO constructed for the problem of identifying the top-$\tau$ highest EC nodes was implemented on the D-Wave 2000Q_LANL machine at Los Alamos National Laboratory [45] and also on IBM-Q using `IBM Qiskit QASM` simulator or real quantum devices available on the IBM-Q, in particular `ibmq_manhattan` [42]. At the front-end of the D-Wave platform, we use `D-Wave Ocean` tools to submit instructions for the optimal ground state solution for the problem Hamiltonian/QUBO with specified parameters such as the anneal time, chain strength, post-processing method and the number of samples to be collected. The front-end then sends the instructions to the

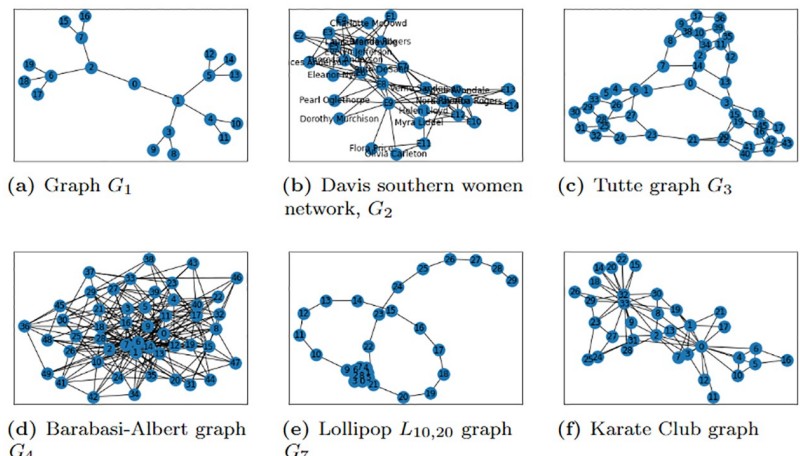

**Fig 2. Large graphs considered.** (a) Graph $G_1$, (b) Davis southern women network, $G_2$ (c) Tutte graph $G_3$, (d) Barabasi-Albert graph $G_4$, (e) Lollipop $L_{10,20}$ graph $G_7$, (f) Karate Club graph.

2000Q_LANL solver chip for processing. Once the problem Hamiltonian is successfully embedded unto the chip, the annealer solves the QUBO for the minimum energy solution which is a bit string that minimally violates the constraint. Note that the bit string returned has $\tau$ number of 1's whose corresponding index denote the the top $\tau$ influential nodes of the graph. To solve the QUBO problem on the IBM-Q Experience platform, we employed Qiskit's CPLEX tools [46] in generating a quadratic program that is converted to a QUBO/Ising operator for building quantum instances on available QASM simulators or real quantum devices available on the IBM-Q. The ground state of the QUBO Hamiltonian is then solved using a Minimum Eigen Solver [47] such as quantum approximate optimization algorithm (QAOA) [48]. Due to qubit limitation, the QUBO can be implemented for graphs with at most 65 nodes on IBM-Q's Manhattan which can encode at most 65 qubits.

## Results

Our investigation considered fabricated synthetic and non-synthetic graphs such as shown in Figs 1 and 2. These graphs were created using the NetworkX graph generator algorithms [49]. The graphs in Fig 2 include scale-free networks, that is networks with power law or scale-free degree distribution. The Barabasi-Albert graph, $G_4$ (see Fig 2d), is an example of a scale-free network which integrates two essential concepts in real networks: growth (increasing number of nodes in the network) and preferential attachment (highly connected nodes have a maximum likelihood of obtaining new connections) [50]. Other well known graphs considered are social networks such as the Davis Southern women social network (see Fig 2b)—the network of a Southern women social club made up of 18 women who attended 14 different events, and the Karate Club graph (see Fig 2f)—network of a university karate club, Bull graph, $G_{10}$ (see Fig 1d), complete graphs $K_n$ (e.g., see Fig 1f), complete bipartite graphs $K_{m,n}$ (e.g., see Fig 1e), Sedgewick-Maze graph (see Fig 1a), Barbell graph—two complete graphs joined together by a path graph, (e.g., the Lollipop graph, $G_7$ (see Fig 2f)), Tutte graph $G_3$ (see Fig 2c)—a cubic polyhedral graph with 46 nodes and 69 edges. The fabricated graphs were mostly tree graphs– connected acyclic undirected graphs such as graphs $G_1$ (see Fig 2a), and $G_8$ (see Fig 2b). The tree graphs $G_1$ and $G_8$ were fabricated mainly to test and show that the QUBO formulation is correctly identifying the top-$\tau$ most important nodes based on EC rather than a degree centrality measure.

NetworkX was used to obtain initial EC measures and node rankings for comparison with results from the quantum computations on small graphs.

## Discussion

Encoding the QUBO problem P on both the D-Wave 2000Q and IBM-Q devices (QASM simulator and ibmq_manhattan for the Karate club graph), we obtained results that showed the ground state of the QUBO $Q$ identifies the top-1 highest EC node when compared to the results from the power iterative EC method of NetworkX.

Fig 3b shows the graph of the output for a search for the top-1 most influential node (colored yellow) of the graph $G_8$ in Fig 1. Comparing the two graphs in Fig 3a and 3b, we observe that the quantum computing scheme correctly identifies the node with top-1 highest EC value and not that of high degree centrality value. We probed the problem further with different values of $\tau \geq 1$ and compared the results with that of NetworkX. Fig 3c, shows the results obtained for a search for the top-($\tau = 5$) most influential nodes (in yellow) of the graph $G_8$. For the graph of the NetworkX results, the most central nodes are identified by the size and brightness of the color of the disk around the nodes; the larger and brighter the disk the more central the node. For the graph $G_8$, the yellow node (0) is the most central, followed by nodes 1, 2, 3

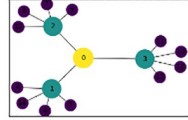

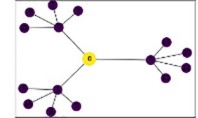

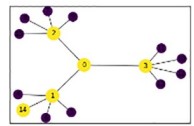

(a) NetworkX ranking

(b) Ranking obtained from IBM-Q and D-Wave for $\tau = 1$

(c) Ranking obtained on both IBM-Q and D-Wave using $\tau = 5$

**Fig 3. Most central nodes using QUBO and NetworkX's EC.** The graph $G_8$ showing the most central nodes using the (a) NetworkX EC algorithm, and QUBO with (b) $\tau = 1$ and (c) $\tau = 5$ on IBM-Q and D-Wave quantum computers for $P_0 = \frac{1}{\sqrt{n}}$ and $P_1 = 5n$ where $n$ is the number of nodes of the graphs. For (a), the most central node(s) is(are) of brighter colors and are encircled by larger circles. In (b) and (c), the bright colored (yellow) nodes are the most central nodes whiles the dark colored (purple) nodes are the least central nodes relative to $\tau$.

and the nodes 4, 5, 6, . . ., 14 are all of the same centrality value and are the least central nodes. It was observed that it is sufficient to choose the penalty constants $P_0 = \frac{1}{\sqrt{n}}$, and $P_1 > P_0$ (in our case $P_1 = 5n$ worked well). The value $P_0 = \frac{1}{\sqrt{n}}$ is chosen because the value $\gamma = \frac{1}{\sqrt{n}}$ was found to be the optimal value in the quantum search by continuous time quantum walk [38].

To obtain optimal results using the D-Wave 2000Q, it is best to set the chain strength to the maximum possible, 1000 in this case. It was observed that very low chain strength values resulted in broken chains affecting the probability of the QA settling into a global minimum solution. Setting the post-processing method to "optimization" and the number of samples to the maximum, 10, 000 boosted the chances of obtaining the optimal solution. However as the graph gets larger more samples are required to increase the probability of observing the ground state solution. Here is where some inconsistencies in obtaining the lowest energy output for larger graphs was highlighted: the first (or second, etc.) run may not result in the expected output. Running multiple times would eventually result in the correct output occurring once, but due to the noisy and quantum nature of the quantum machine, there is no fixed number of runs determined for all graphs to guarantee the global minimum solution. This was mostly observed in the Karate Club graph (graph with 34 nodes) in Fig 4e and the Davis Southern women network. With a little bit of luck the result can be obtained in the first run or in a few runs. For example, in Fig 5 we see that QA returns the global minimum for the QUBO problem P with $\tau = 3$ for the graphs with 34 and 50 nodes after 5 and 2 runs respectively and returns the global minimum for the QUBO for other graphs after 1 run. This behavior is not surprising since an increase in the problem size decreases the probability of finding an optimal solution due to annealing error and imperfect hardware [51].

The quadratic solver QBSolv in Ocean and the exact classical Numpy Minimum Eigen Solver in Qiskit served as a benchmark for determining the correct global minimum of the D-Wave annealing output and the QAOA output from IBM-Q respectively. QBSolv provided expected results of the minimization on most occasions with degenerate ground state solutions in some instances. Degeneracy here refers to the outputs with the same minimum energy value of the QUBO due to multiple nodes having the same centrality value. Whenever there is degeneracy in the QBSolv output, one of the solutions correctly identifies the top-$\tau$ highest EC nodes. For example, in Fig 3c, we have 12 degenerate solutions corresponding to the 12 leaves of the tree graph. The solution graphed included node 14 in the top 5 important nodes. However, node 4, 5, 6, 7, 8, 9, 10, 11, 12, 13 or 15 are valid replacements for node 14. In this output, all these nodes have the same centrality values (see Table 1 for example). The minimum solution obtained is affirmed global minimum when it's energy equals that of the exact classical

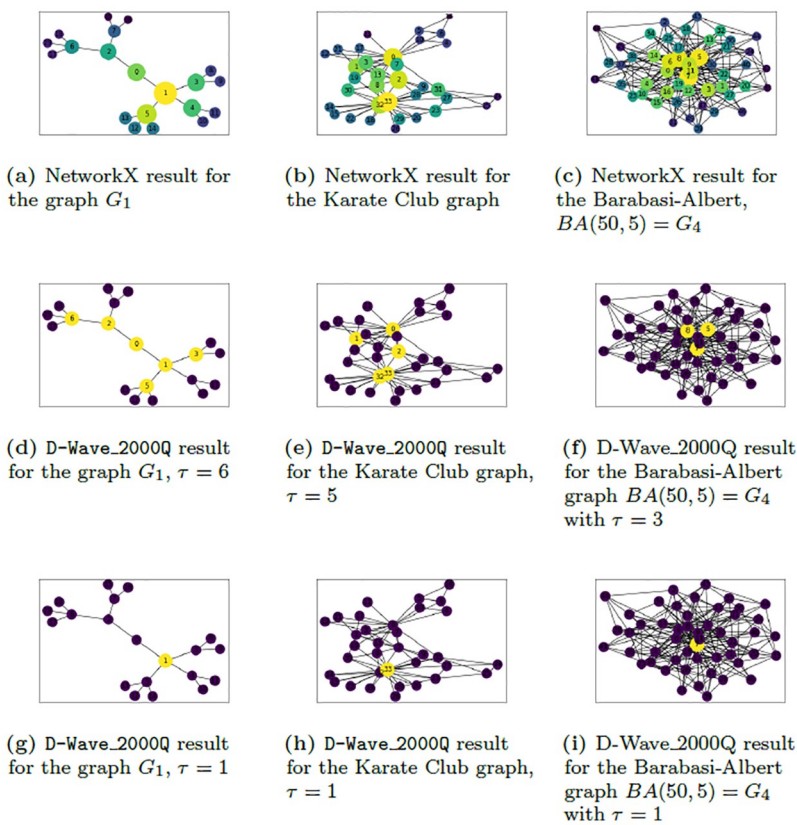

**Fig 4. Results for challenging graphs.** NetworkX results for the challenging graphs encountered; (a) graph $G_1$, (b) Karate Club graph, and (c) Barabasi-Albert graph, $BA(50, 5) = G_4$. D-Wave results obtained for the QUBO in Eq 11 using $P_0 = \frac{1}{\sqrt{n}}$ and $P_1 = 5n$ for (d) graph $G_1$ with $\tau = 6$, (e) Karate Club graph with $\tau = 5$, (f) $BA(50, 5) = G_4$ with $\tau = 3$, (g) $G_1$ with $\tau = 1$, (h) Karate Club graph with $\tau = 1$ and (i) $BA(50, 5) = G_4$ with $\tau = 1$. The D-Wave results colors the top $\tau$ most important nodes yellow and the least central nodes purple. The NetworkX result identifies most central nodes with larger and brighter circles (yellow being the top) and least central nodes with smaller and darker circles (least being purple).

Numpy Minimum Eigensolver algorithm (for implementations on IBM-Q) or tabu QBSolv (for implementations on D-Wave 2000Q).

Solving the problem for the QUBO in Eq (11) for the graph in Fig 6 and graph $G_8$ in Fig 2 correctly identify the most important nodes for any given $\tau$, including selecting node 0 as the highest EC node (see Fig 3). The penalty weights used here were $P_0 = \frac{1}{\sqrt{n}}$ and $P_1 = 5n$ for $n$ number of nodes of the graphs. The same results were obtained from experiments on IBM-Q's QASM simulator using QAOA. However the ground state solution for the graphs with smaller numbers of nodes ($n \leq 16$) were obtained on a single run with the penalty constants $P_0 = \frac{1}{\sqrt{n}}$ and $P_1 = 5n$. When considering graphs with larger numbers of nodes, the program had to be run multiple times using the same penalty constants above to be able to capture the global minimum solution.

Solving for the ground state solution of the QUBO in Eq (11) for the graph $G_1$ and Karate Club graph in Fig 2 was quite challenging. On most occasions, the solution for the Karate Club graph with $\tau \geq 3$ required multiple runs before settling on the global minimum solution. In other words, this graph required more samples to be able to output the global minimum. For $3 \leq \tau \leq 5$, the QUBO couldn't capture the ordering that matched that of NetworkX rankings when using the penalty constants $P_0 = \frac{1}{\sqrt{n}}$ and $P_1 = 5n$ for the graph $G_1$. The nodes 0, 3, 4 were

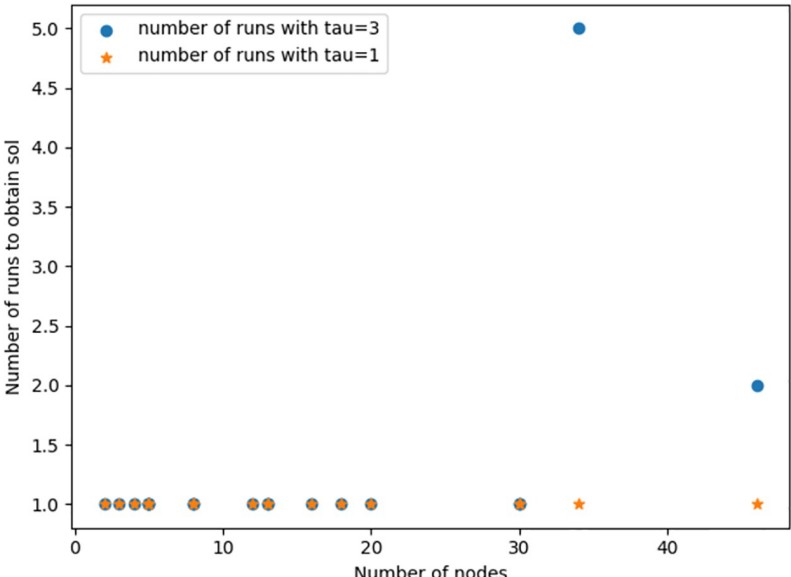

**Fig 5. Number of runs plot ($\tau = 1$ and $\tau = 3$).** A plot of the number of nodes against the number of runs to obtain optimal solution to the QUBO problem P, on the D-Wave 2000Q.

**Table 1. Table of Networkx and QUBO results with $\tau = 1$ and $\tau = 5$.**

| Graph $G = (V, E)$ Features | | | | Most central node by EC | | Top 5 most central nodes | |
|---|---|---|---|---|---|---|---|
| Name | \|V\| | \|E\| | Density $\rho(G)$ | Nx | QA | Nx | QA |
| Bull | 5 | 5 | 0.5 | 1 | 1 | 0,1,2,3,4 | 0,1,2,3,4 |
| $K_{3,5}$ | 8 | 15 | 0.54 | 0,1,2 | 2 | 0,1,2,3,4 | 0,1,2,3,4 |
| $K_5$ | 5 | 10 | 1 | - | 0 | - | - |
| $K_6 + S_6$ | 12 | 18 | 0.32 | 5 | 5 | 0,1,2,3,4,5 | 0,1,2,3,5 |
| Florentine Family $F$ (Fig 1h) | 15 | 20 | 0.19 | Medici | Medici | Guadagni, Medici, Ridolfi, Strozzi, Tomabuoni | Guadagni, Medici, Ridolfi, Strozzi, Tomabuoni |
| Tutte $G_3$ | 46 | 69 | 0.07 | - | 7 | - | 12, 30, 36, 44, 45 |
| Lollipop $L_{10,20} = G_7$ | 30 | 65 | 0.15 | 9 | 9 | 0,1,2,4,9 | 0,1,2,4,9 |
| Barabasi-Albert $BA(50, 5, 7)$ | 50 | 225 | 0.18 | 7 | 7 | 3,5,6,7,8 | 3,5,6,7,8 |
| Davis southern women $G_2$ | 32 | 89 | 0.18 | E8 | E8 | E7,E8,E9, Evelyn Jefferson, Theresa Anderson | E7,E8,E9, Evelyn Jefferson, Theresa Anderson |
| Karate club | 34 | 78 | 0.14 | 33 | 33 | 0,1,2,32,33 | 0,1,2,32,33 |
| Sedgewick-Maze | 8 | 10 | 0.36 | 4 | 4 | 0,3,4,5,7 | 0,3,4,5,7 |
| $G_8$ | 16 | 15 | 0.13 | 0 | 0 | 0,1,2,3,4 | 0,1,2,3,8 |
| $G_1$ | 20 | 19 | 0.1 | 1 | 1 | 0,1,2,3,5 | 0,1,2,5,6 |
| $M$ (Fig 1i) | 13 | 15 | 0.19 | 2 | 2 | 0,1,2,6,9 | 0,1,2,6,9 |

The first 4 columns describe some basic graph features such as the name of the graph, number of nodes $|V|$, number of edges $|E|$ and the density of the graph $\rho = \frac{2|E|}{|V|(|V|-1)}$. Columns 5 and 6 describe the most central node corresponding to the EC NetworkX result and D-Wave QA result for the QUBO. The last two columns describe the top 5 most central nodes using the EC measure in NetworkX and by implementing the QUBO, $Q$, on D-Wave (QA). NB: The QA result for the top 5 most central nodes of the BA graph was obtained by applying QA followed by a greedy steepest descent post-processing method. A"-" signifies no node was identified as most central or least central, all nodes have the same centrality.

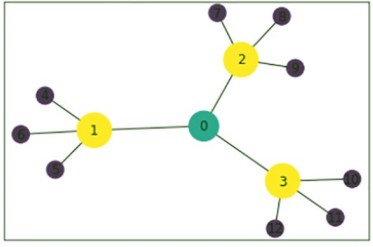 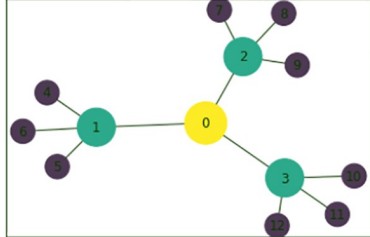

(a) Degree centrality    (b) Eigenvector centrality

**Fig 6. Degree and eigenvector centrality of a Graph.** Identifying the importance of a node in a network based on (a) degree centrality and (b) eigenvector centrality. The color and radius of each disk around a node is dependent upon the centrality values. The least central nodes are colored in purple, the mid-central nodes are colored in green with the most central nodes colored yellow. Nodes 1, 2 and 3 have the highest values when using the degree centrality measure while node 0 has the highest value when using the EC measure.

always skipped in the search for the top τ = 3, 4, 5 highest EC nodes. From Fig 4d, we see that the output for the top 6 most important nodes excludes node 4 and includes node 2 which shouldn't be the case since from Fig 4a, node 4 is more central than node 2. The exact reason for this occurrence for this particular graph is unknown, however it seems the program picks only one of the degenerate nodes 3, or 4 and moves on to select the next central node 2.

With these interesting results, we further examined the possibility of defining a hierarchy of nodes in the network from the QUBO results obtained from the D-Wave 2000Q and IBM-Q. By hierarchy we mean a ranking that orders the nodes based on importance or influence using EC. The hierarchy can then be used to identify for example super-spreaders of disease. We compare the hierarchy of nodes obtained using our QUBO formulation with that obtained from NetworkX when using the power iterative method of EC algorithm. The result obtained for graph M is shown in Table 2. To determine the node rank, we consider the set of τ nodes

**Table 2. Node rank of graph M using QUBO results.**

| τ | (QA/QAOA) QUBO results for top τ most central nodes of graph M | Node Rank of (QA/QAOA) QUBO results for M | Node Rank of NetworkX EC result for M |
|---|---|---|---|
| 1 | 2 | 2 | 2 |
| 2 | 0, 2 | 0 | 0 |
| 3 | 0, 2, 9 | 9 | 9 |
| 4 | 0, 1, 2, 9 | 1 | 1 |
| 5 | 0, 1, 2, 6, 9 | 6 | 6 |
| 6 | 0, 1, 2, 3, 6, 9 | 3 | 3 |
| 7 | 0, 1, 2, 3, 5, 6, 9 | 5 | 5 |
| 8 | 0, 1, 2, 3, 5, 6, 9, 11 | 11 | 11 |
| 9 | 0, 1, 2, 3, 4, 5, 6, 9, 11 | 4 | 7 |
| 10 | 0, 1, 2, 3, 4, 5, 6, 7, 9, 11 | 7 | 8 |
| 11 | 0, 1, 2, 3, 4, 5, 6, 7, 8, 9, 11 | 8 | 4 |
| 12 | 0, 1, 2, 3, 4, 5, 6, 7, 8, 9, 11, 12 | 12 | 10 |
| 13 | 0, 1, 2, 3, 4, 5, 6, 7, 8, 9, 10, 11, 12 | 10 | 12 |

Determining node rank from the QUBO result obtained from D-Wave (QA) and IBM-Q (QAOA) using symmetric difference of results for τ = 1 to τ = n. In column 4, the ith in rank is determined by taking the difference of the result for τ = i and τ = i − 1 e.g. the 1st (most central) node 2 is determined by implementing the QUBO for τ = 1, the 2nd is determined by taking the difference between the result of τ = 2, {0, 2} and the result of τ = 1, {2}, i.e. {0, 2}\{2} = {0}, implying node 0 is ranked second in importance.

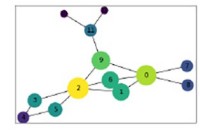
(a) NetworkX ranking

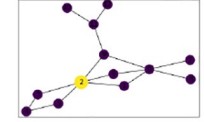
(b) Ranking obtained from IBM-Q and D-Wave for $\tau = 1$

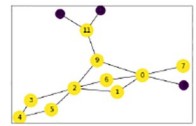
(c) Ranking obtained from IBM-Q and D-Wave using $\tau = 10$

**Fig 7. Results for graph M.** The graph $M$ showing the most central nodes using (a) the NetworkX EC algorithm, and ((b) and (C)) QUBO on the IBM-Q and D-Wave quantum computers for (b) $\tau = 1$ and (c) $\tau = 10$ with $P_0 = \frac{1}{\sqrt{n}}$ and $P_1$ = 5$n$ where $n$ is the number of nodes of the graphs. For (a), the most central node(s) is(are) of brighter colors and are encircled by larger circles. In (b) and (c), the bright colored (yellow) nodes are the most central nodes whiles the dark colored (purple) nodes are the least central nodes relative to $\tau$.

obtained from the QUBO results for each value of $0 < \tau \leq n$ and compute the symmetric difference. The $i$th rank is determined by taking the difference of the QUBO result for $\tau = i$ and $\tau = i - 1$. For example, to rank the nodes for graph $M$, the 1st most important node is obtained by running the QUBO for $\tau = 1$ (see Fig 7b). The 2nd most important node is determined by taking the difference between the QUBO results for $\tau = 2$, {0, 2} and the result {2} of $\tau = 1$. Thus {0, 2} \ {2} = {0} implies that node 0 is the second most important node. Continuing in this manner the importance of the nodes can be ranked. Computing the difference {0, 1, 2, 3, 4, 5, 6, 7, 9, 11} \ {0, 1, 2, 3, 4, 5, 6, 9, 11} = {7} for $\tau = 10$ and $\tau = 9$ we see that the 10th important node is node 7.

An examination of the computational timings of the results from both noise-free and noise-model simulators favored the classical solvers, Numpy Minimum Eigensolver and tabu qbsolv, over quantum solvers, QA and QAOA, in generating results for the QUBO problem for $n \leq 50$ (see Fig 8). The time plots in Fig 8 show that as the number of nodes ($n$) increases, the total wall clock time spent by the quantum simulators to return a solution increases rapidly

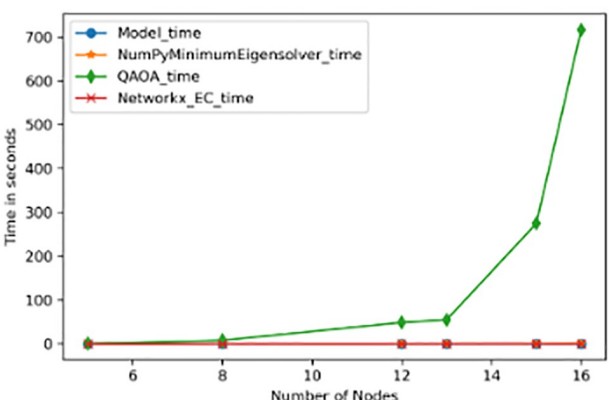

(a) Times from IBM-Q ($\tau = 1$)

(b) Times from D-Wave 2000Q ($\tau = 1$ and $\tau = 3$)

**Fig 8. Time plots.** A plot of the number of nodes against the time in seconds taken to solve the QUBO problem P on (a) the IBM-Q using (1) the classical exact numpy minimum eigensolver, (2) the QAOA algorithm on a QASM simulator, and using the power iterative method in NetworkX to solve the *EC* problem, and on (b) the D-Wave 2000Q using (1) the classical qbsolv solver which employs the tabu algorithm on the D-Wave system, (2) the QA algorithm and using the power iterative method in NetworkX to solve the *EC* problem.

compared to the time spent by the classical solvers and the power iterative method employed by NetworkX. The service time is the total wall clock time for a quantum machine instruction (QMI) to be sent to the D-Wave system, execute on the D-Wave system, and return a solution. It can be obtained by computing the time difference between the *start time* and *end time* for each QMI [52]. QPU access time is the time to execute a single QMI on the QPU and is composed of the QPU programming and sampling time. QPU programming time is basically a measure of the time taken to initialize the problem onto the QPU whereas the QPU sampling time is typically a measure of the time taken to actually execute the problem on the QPU. The QPU sampling time is made up of multiple anneal and readout times (taken to perform and read samples from the QPU) as well as the thermalization (time taken for the QPU to recover it's initial temperature) [52, 53]. The QAOA time in Fig 8a is calculated here as the total time taken to build (model time) and submit the QUBO to the IBM-Q QASM simulator plus the time taken to receive a solution.

Error-mitigation techniques exist and can be applied to improve the results whenever there is an overwhelming noise affecting the performance. One of the methods we employed is the extended J range which involves the setting of a strong chain strength in defining a minor embedding [54]. This technique was employed to prevent broken chains and their negative effect on the D-Wave output. For our problem, low chain strength implied more broken chains. However after increasing the chain strength to the maximum (1000) at the time, we were able to obtain the desired global optima. Other techniques used include increasing the number of reads, annealing time and number of samples which have a positive influence on the probability of obtaining the optimal solution [55]. Another error-mitigation method is increasing the spin-reversal (Gauge) transforms, a technique that controls the influence of biases with a side effect of an increase in total run time of the problem [55]. By default, the D-Wave system also uses a drift correction technique every hour to correct drifts [56]. The graph $BA(50, 5)$ is the largest random graph considered on the D-Wave system. It is a dense graph and as $\tau$ increases the QUBO, $Q$ becomes more dense and challenging to handle on the D-Wave quantum annealer. We observed no match between the global minimum energy of the Tabu QBSolv result and the minimum energy of QA result after 100 runs of the problem P with $\tau = 5$ on D-Wave 2000Q. However, after performing a greedy steepest descent post-processing on the QA solution resulted in a solution that matched the Tabu QBSolv solution and that worked usually on the 1st run. This is not surprising as we recall that D-Wave results are often approximate solutions and not always exact but can be improved by following it with post-processing methods (in this special case, running a greedy steepest descent was helpful [57]). It is interesting to note that solving this same problem with simulated annealing yielded the result after 1 run.

## Conclusion

We have formulated and empirically shown the ground state of the QUBO problem P identifies the top most important node in a graph based on the EC measure. Using quantum computing algorithms such as quantum annealing on the D-Wave 2000Q and QAOA on IBM-Q, our formulation was able to correctly identify all top-$(\tau < n)$ most important nodes for graphs with less than 17 nodes ($n < 17$). For graphs with more than 16 nodes, the quantum computing algorithm always identified the top $\tau \leq 6$ most important nodes for all the graphs considered correctly except for the Davis Southern women network and the tree graph $G_1$ whose outputs for some values of $1 < \tau \leq 6$ showed some marginal inconsistencies. Marginal because the differences are negligible. For example, for the graph $G_1$, the node 3 was not selected since it's centrality value is the same as that of node 4. Despite this challenge, the results obtained

from all graphs considered for $\tau = 1$ support the claim that the ground state of the QUBO problem P identifies the top-1 highest EC node. We have also demonstrated, although using a complex approach, the feasibility of defining a hierarchy of nodes in a graph from our formulation using the QUBO results from the D-Wave 2000Q and IBM-Q's QASM simulator. Given that the current quantum resources (D-Wave 2000Q and IBM-Q's Manhattan) at our disposal limit us on the size of graphs to explore and in the presence of uncontrollable noise which affects the probabilities of obtaining quality results, we were unable to experiment with real life data. Therefore in the future when powerful and less noisy quantum computers are made available for our perusal, we wish to test our hypothesis further to establish a more generalized formulation that works for all $\tau$ on all graphs. That is, we want to verify.

**Claim 1** *For any graph G = (V, E), with adjacency matrix A and degree sequence $\boldsymbol{d}$. The indices of the nonzero elements of the ground state solution to the QUBO problem*

$$\min_{\boldsymbol{x} \in \{0,1\}^n} \boldsymbol{x}^T Q \boldsymbol{x}$$

$$Q = -P_0 A^2 \hat{\boldsymbol{d}} \hat{\boldsymbol{d}}^T A - P_0 A \hat{\boldsymbol{d}} \hat{\boldsymbol{d}}^T A^2 + P_1 C$$

$$C = (1 - 2\tau)I + U$$

*where the matrix U is defined in* Eq (9), *corresponds to the $\tau$ most central nodes via EC measure of the graph G for any $\tau \leq n = |V|$ and $P_1 > P_0 \neq 0$.*

For this claim we wish to investigate both directed and undirected graphs, "will replacing $A^2$ by $AA^T$ or $A^T A$ in Q still work for directed graphs or will it require a modified QUBO?"

## Supporting information

**S1 Appendix. Computing degree centrality and eigencentrality from exponential function [10].**
(PDF)

## Acknowledgments

We acknowledge the ASC program at LANL for use of their Ising D-Wave 2000Q quantum computing resource. We also acknowledge the use of the D-Wave Leap 2000Q quantum computing resource. Quantum resources from the IBM-Q Hub are also acknowledged. Assigned: Los Alamos Unclassified Report LA-UR-21-24030.

## Author Contributions

**Conceptualization:** Christian F. A. Negre, Susan M. Mniszewski.

**Formal analysis:** Prosper D. Akrobotu, Tamsin E. James, Christian F. A. Negre.

**Funding acquisition:** Christian F. A. Negre, Susan M. Mniszewski.

**Methodology:** Prosper D. Akrobotu, Tamsin E. James, Christian F. A. Negre, Susan M. Mniszewski.

**Software:** Prosper D. Akrobotu, Tamsin E. James, Christian F. A. Negre, Susan M. Mniszewski.

**Supervision:** Christian F. A. Negre, Susan M. Mniszewski.

**Validation:** Prosper D. Akrobotu, Tamsin E. James.

**Writing – original draft:** Prosper D. Akrobotu, Tamsin E. James.

**Writing – review & editing:** Prosper D. Akrobotu, Tamsin E. James, Christian F. A. Negre, Susan M. Mniszewski.

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
