## [Decision Letter · Decision Letter 0]

11 Apr 2022

PONE-D-21-32688A QUBO Formulation for EigencentralityPLOS ONE

Dear Dr. Akrobotu,

Thank you for submitting your manuscript to PLOS ONE. After careful consideration, we feel that it has merit but does not fully meet PLOS ONE’s publication criteria as it currently stands. Therefore, we invite you to submit a revised version of the manuscript that addresses the points raised during the review process.

We look forward to receiving your revised manuscript.

Kind regards,

Gábor Vattay, PhD, DSc

Academic Editor

PLOS ONE

“Research presented in this article was supported by the Laboratory Directed Research and Development (LDRD) program of Los Alamos National Laboratory (LANL) under project number 20200056DR. LANL is operated by Triad National Security, LLC, for the National Nuclear Security Administration (NNSA) of the U. S. Department of Energy (contract no. 89233218CNA000001). TJ acknowledges support from the U.S. Department of Energy (DOE) through a quantum computing program sponsored by the Los Alamos National Laboratory (LANL) Information Science & Technology Institute. This research was also supported by the U. S. Department of Energy (DOE) National Nuclear Security Administration (NNSA) Advanced Simulation and Computing (ASC) program at LANL. We acknowledge the ASC program at LANL for use of their Ising D-Wave 2000Q quantum computing resource. Quantum resources from the IBM-Q Hub are also acknowledged. Assigned: Los Alamos Unclassified Report LA-UR-21-24030.

“This research was supported by the U.S. Department of Energy (DOE) National Nuclear Security

Administration (NNSA) Advanced Simulation and Computing (ASC) program at Los Alamos

National Laboratory (LANL). This research has been funded by the LANL Laboratory Directed

Research and Development (LDRD) and ASC program. SMM, CFAN, and PDA were funded by

LANL LDRD. TEJ was funded by the U.S. Department of Energy (DOE) through a quantum

computing program sponsored by the Los Alamos National Laboratory (LANL) Information

Science & Technology Institute. Assigned: Los Alamos Unclassified Report LA-UR-21-24030.

LANL is operated by Triad National Security, LLC, for the National Nuclear Security

Administration of U.S. Department of Energy (Contract No. 89233218NCA000001).  http://www.lanl.gov

Reviewers' comments:

Reviewer's Responses to Questions

**Comments to the Author**

1. Is the manuscript technically sound, and do the data support the conclusions?

Reviewer #1: Partly

Reviewer #2: Yes

2. Has the statistical analysis been performed appropriately and rigorously? 

Reviewer #1: N/A

Reviewer #2: Yes

3. Have the authors made all data underlying the findings in their manuscript fully available?

Reviewer #1: Yes

Reviewer #2: No

4. Is the manuscript presented in an intelligible fashion and written in standard English?

Reviewer #1: Yes

Reviewer #2: Yes

5. Review Comments to the Author

Reviewer #1: The paper presents an approach for identifying a given number of nodes with the highest eigencentrality scores using quantum hardware. The paper presents a QUBO formulation of the problem that can be solved on quantum hardware such as quantum annealers and gate-based quantum computers. Experimental analysis shows the proposed approach manages to find the most central node and the top 5 most central nodes according to eigencentrality scores, as well as correctly rank all nodes for graphs up to 16 nodes.

The paper is interesting and deals with the important problem of identifying the most central nodes in a network. I think the core contribution presented in the work (a QUBO formulation for identifying a given number of nodes with the highest eigencentrality scores) is important and the experiments indicate that the approach performs well (on small graphs) when compared to the baseline classical approach. However, I think the current manuscript has some major issues that need to be addressed.

1. Claims are not clear enough

- What exactly is the problem being solved? The text, in some places, seems to focus on the problem of computing eigencentrality scores or at least is not clear enough on what the problem being solved. For example, "we lay the foundation for the calculation of eigenvector centrality using quantum ...." or "Eigenvector centrality as a QUBO problem" is different than a formulation that identify the top N most central users. While it can be run repeatedly to generate ranking, this is a much more complex procedure, and still does not actually compute the scores but the ranking. This seems to also impact the presentation of the approach: instead of a clear problem definition, the paper presents the original problem of eigencentrality calculation and transforms it into a different problem that can be formulated using QUBO. I think the paper would benefit from a clear problem definition (i.e., identifying a given number of most central users) and then a QUBO formulation that solves this problem.

- As there is no clear problem definition, the paper also does not provide a clear statement on the connection between the proposed solution and the problem being solved. For example, it is not clear if the optimal solution for the QUBO formulation (assuming it is found) is guaranteed to be the optimal solution for the problem (i.e., the given number of most central nodes) or is a heuristic approach that seems to work well in practice (see related points on Proposition 1 and Claim 1 below).

2. No sufficient theoretical support for proposed approach and claims:

- No proof for proposition 1, and there isn't a more general proof for the case of \\tau > 1 (which seems to be formulated in Claim 1). The paper mentions the authors were able to "verify for proposition 1 (with \\tau >= 1) for ..." however these are empirical results and not a proof.

- There are no theoretical results on sufficient/optimal selection of P values (the Lagrange multiplier). This is also related to the previous point as I was surprised that proposition 1 does not include any conditions on the P values (e.g., is it still correct when P_0 or P_1 are equal to zero?). Also, there is no principled explanation/justification for the values used in the experiments.

3. The empirical support could be strengthened and some important details on the experiments should be clarified:

- As mentioned earlier, many results are only verified experimentally and not theoretically. Since all the graphs used are very small and in a limited range of values, it is not clear how the results will change as graphs become larger (e.g., should the P_0 and P_1 values used remain the same?). While the limited size is justified by the hardware limitations, larger graphs could potentially be tested on simulators.

- The paper mentions that in some graphs there was a need for multiple runs to capture global minimum. How many runs were used and how was it determined if the global minimum was captured or additional runs are needed?

- Why is there no result (or "-") in the "top 5 most central nodes" for QA in the BA(50,5) graph?

- The paper should list and cite the concrete algorithms being used and not just the libraries, e.g., which algorithm (that is implemented in NetworkX) is used for computing EC? which classical optimization algorithm (that is implemented in SciPy) was used?

- It was not clear to me what claim is supported by the analysis of the performance of SciPy in Table 2 (related to point (1) above)

Minor points:

- Text refers to section numbers (e.g., "Section 2") but the sections do not have numbers

- Figures are listed in the paper but appear at the end

- First sentence in Conclusion is not clear: "We have formulated and shown a QUBO problem for .... is possible".

Reviewer #2: This is an interesting work. Honestly, I am not familiar to the topic of centrality and its application in graph. In this sense, the manuscript gives a brief and sound overview. The structure of the paper looks reasonable. My only comment is that there is a lack of the discussion regarding the noise during the simulations. A suggested revision can be focused on (1) presenting/comparing the data/results from both noise-free and noise-model simulator, as well as how both of them are compared to the data from real devices, and (2) if the noise is overwhelming and altering the some observations, then is there any error-mitigation can be applied to improve the results.

6. PLOS authors have the option to publish the peer review history of their article (what does this mean?). If published, this will include your full peer review and any attached files.

Reviewer #1: No

Reviewer #2: No

---

## [Author Response · Author response to Decision Letter 0]

10 Jun 2022

We are thankful to the anonymous reviewers for their helpful comments and have prepared a detailed response in the pdf, Response to Reviewers.pdf. Each referee's comments are numbered and quoted in bold italic typeface, our replies can be found below each comment in roman typeface. New sections added to the manuscript are written in red italic typeface.

---

## [Decision Letter · Decision Letter 1]

28 Jun 2022

A QUBO formulation for top-τ eigencentrality nodes

PONE-D-21-32688R1

Dear Dr. Akrobotu,

We’re pleased to inform you that your manuscript has been judged scientifically suitable for publication and will be formally accepted for publication once it meets all outstanding technical requirements.

Kind regards,

Gábor Vattay, PhD, DSc

Academic Editor

PLOS ONE

Additional Editor Comments (optional):

Reviewers' comments:

Reviewer's Responses to Questions

**Comments to the Author**

1. If the authors have adequately addressed your comments raised in a previous round of review and you feel that this manuscript is now acceptable for publication, you may indicate that here to bypass the “Comments to the Author” section, enter your conflict of interest statement in the “Confidential to Editor” section, and submit your "Accept" recommendation.

Reviewer #1: All comments have been addressed

Reviewer #2: All comments have been addressed

2. Is the manuscript technically sound, and do the data support the conclusions?

Reviewer #1: Yes

Reviewer #2: Yes

3. Has the statistical analysis been performed appropriately and rigorously? 

Reviewer #1: N/A

Reviewer #2: Yes

4. Have the authors made all data underlying the findings in their manuscript fully available?

Reviewer #1: Yes

Reviewer #2: Yes

5. Is the manuscript presented in an intelligible fashion and written in standard English?

Reviewer #1: Yes

Reviewer #2: Yes

6. Review Comments to the Author

Reviewer #1: Minor comment:

- It is not entirely clear to me from the authors' answers if the results reported (e.g., in Table 1) are after running greedy steepest descent. If that is the case, it should be clearly stated before the results are presented.

Reviewer #2: All my comments have been addressed by authors' revision. I thus have no further comments, and recommend its publication in PLOS ONE

7. PLOS authors have the option to publish the peer review history of their article (what does this mean?). If published, this will include your full peer review and any attached files.

Reviewer #1: No

Reviewer #2: No

---

## [Editor Report · Acceptance letter]

4 Jul 2022

PONE-D-21-32688R1 

A QUBO formulation for top-τ eigencentrality nodes 

Dear Dr. Akrobotu:

I'm pleased to inform you that your manuscript has been deemed suitable for publication in PLOS ONE. Congratulations! Your manuscript is now with our production department. 

Kind regards, 

on behalf of

Dr. Gábor Vattay 

Academic Editor

PLOS ONE